# Comparison of Laparoscopic and Open Radical Cystectomy for Muscle-Invasive Bladder Cancer

**DOI:** 10.3390/ijerph192315995

**Published:** 2022-11-30

**Authors:** Janusz Lisiński, Jakub Kienitz, Piotr Tousty, Krystian Kaczmarek, Artur Lemiński, Marcin Słojewski

**Affiliations:** 1Department of Urology and Urological Oncology, Pomeranian Medical University, Powstańców Wielkopolskich 72, 70-111 Szczecin, Poland; 2Department of Obstetrics and Gynecology, Pomeranian Medical University, Powstańców Wielkopolskich 72, 70-111 Szczecin, Poland

**Keywords:** bladder cancer, complications, laparoscopy, radical cystectomy

## Abstract

The goal of the study was to compare laparoscopic and open radical cystectomy in treatment of muscle-invasive bladder cancer in the Department of Urology and Oncological Urology PUM in Szczecin. A total of 78 patients in the study group underwent laparoscopic cystectomy between 2016–2018, and 81 patients from the control group had open cystectomy between 2014–2016. Both groups were comparable in terms of age, stage, and concomitant diseases. The 3 year overall survival was comparable in both groups (37.7% for laparoscopy and 44.4% for open, *p* = 0.64). There was no difference in positive surgical margin rate. Lymph node yield during cystectomy was higher in open cystectomy (14 vs. 11.5, *p* = 0.001). Post-operative blood loss and transfusion rates were lower in laparoscopic cystectomy. Decrease in hemoglobin level was lower in laparoscopy (0.9 mmol/L, *p* < 0.001). Intraoperative transfusion rate was 11.8% in laparoscopy vs. 34.8% in open cystectomy (*p* = 0.002). Operation time, duration of hospitalisation, and time to full oral alimentation were comparable in both groups. Patients with BMI > 30 kg/m^2^ and those with pT3-T4 cancer in the laparoscopy group had less septic complications post-operatively. Patients with ASA score ≥ 3 from the laparoscopy group had fewer reoperations due to ileus. Laparoscopic cystectomy is less invasive and offers similar oncological outcomes to the open method. Patients benefit from less tissue trauma, less blood loss, and faster recovery. The presented results, as well as other publications, should encourage a wider use of this procedure in everyday urological practice.

## 1. Introduction

Bladder cancer ranks seventh in the global incidence statistics for both genders. In men, it is the fourth most frequently diagnosed malignant neoplasm after prostate, lung, and colon cancer [1]. It is diagnosed in 75% of patients at the local stage, as non-muscle-invasive bladder cancer (NMIBC) [2]. Patients whose cancer initially invades the muscle layer (muscle-invasive bladder cancer (MIBC)) have a much worse prognosis [3]. The main risk factor for bladder cancer development is chronic smoking [4].

Radical cystectomy remains the gold standard in the treatment of patients with MIBC and is one of the options in selected patients with NMIBC [5,6]. The main principles of this operation were described as early as 1949 by Marshall and Whitemore [7]. Despite the passing of the years, the oncological results and the quality of life of the patients undergoing this treatment are still far from expectations [8]. This extensive surgery is usually associated with long hospitalization and convalescence, as well as the risk of complications and death in the perioperative period [9].

The development of laparoscopic techniques has opened up new treatment options for patients with bladder cancer. The early oncological effectiveness of this procedure has been confirmed in prospective studies [10]. Benefits have also been shown in terms of shorter hospitalization, lower blood loss, reduced number of perioperative complications, faster recovery of patients, and higher health-related quality of life. On the other hand, laparoscopic radical cystectomy (LRC) is often associated with a significantly longer operation time [10]. Due to the continuous development of this new treatment method, long-term outcome studies are still awaited.

Our center started laparoscopic treatment of patients with bladder cancer in 2016. The need to properly assess the results of the oncological, functional, and technical aspects of the operation justifies a follow-up examination comparing the new technique with the previously used open radical cystectomy (ORC).

## 2. Materials and Methods

### 2.1. Study and Control Groups

In July 2016, the elective treatment of all patients qualified for radical cystectomy using the laparoscopic method was started in our center. By August 2018, 85 patients had been operated on in this way. They were prospectively included in the analysis as a study group. Patients whose bladder was infiltrated by another tumor, who underwent cystectomy for non-neoplastic reasons, and those who underwent it for urgent indications were excluded from the study Thus, 78 patients were enrolled in the study. Indications for surgery included MIBC, as well as NMIBC for which less invasive treatment (TURB, BCG therapy) was ineffective. All patients operated on at that time in the center qualified for laparoscopic surgery, so there were no specific determining factors about choosing this method. For comparative analysis, a retrospective control group was created that included all consecutive patients who underwent ORC since January 2014. During that time, 89 patients were operated on using this method. The exclusion criteria were the same as for the study group. Thus, 81 patients were enrolled in the study. As in the study group, the indications for the surgery were both MIBC and NMIBC (for which less invasive treatment was not effective).

### 2.2. Group Comparison

The LRC and ORC groups were compared in terms of basic parameters. Gender, age, cancer stage (pathomorphological and clinical), nicotinism, waiting time for the surgery, and presence of hydronephrosis (unilateral or bilateral) were taken into account. Baseline white blood cells (WBC) counts in blood and pre-operative kidney function determined by serum creatinine levels were assessed. The initial nutritional status of the patients was compared based on the body mass index (BMI) and pre-operative albumin and total protein values in the serum. The type of urine drainage was assessed—distinguishing between simple ureterostomy and derivation using the intestine (Bricker and Studer). 

### 2.3. Oncological Results

In order to assess the oncological outcome, overall survival (OS) was assessed, determining the time before death from both oncological and non-cancerous causes. The minimum follow-up period was 36 months for patients operated on in August 2018, the maximum was 78 months for the patients operated on in 2014 (the average observation period was 57 months). Other parameters of the surgery influencing the oncological result were assessed and compared—the number of lymph nodes collected and the presence of a positive surgical margin.

### 2.4. Blood Loss and Transfusion Rate

To objectively assess intraoperative blood loss, the difference in hemoglobin level (Hgb) before and immediately after the surgery was examined. For patients who had intraoperatively transfused red blood cell concentrate (RCC) units, the score was adjusted by 0.62 mmol/L for each unit. Additionally, the maximal decrease in Hgb levels in the first 3 days of postoperative hospitalization was assessed. The transfusion rates of concentrate of packed red blood cells and fresh frozen plasma was compared in both groups.

### 2.5. Complications

The Clavien-Dindo scale was used in both groups to assess post-operative complications. Early complications were classified as those that occurred within 90 days after surgery. Additionally, the following complications were listed, the frequency of which was compared in both groups (Table 3).

### 2.6. Other

Surgery time, duration of hospitalization, duration of antibiotic therapy, and time to full oral nutrition were compared in both groups.

### 2.7. Statistical Analysis

The normality of the distribution of quantitative data was assessed using the Shapiro–Wilk test. The homogeneity of variance was evaluated with Levene’s test. If the assumption of normality of the distribution and homogeneity of variance was met, *t*-test was performed to assess the difference between both groups. In contrast, the Mann–Whitney U test was performed. For qualitative data, the chi-square test with Yates correction was used if required [11]. The Kaplan–Meier estimator was used to estimate patient survival in each group. Survival curves were compared using the log-rank test. The Cox proportional hazards model was used to estimate the effect of selected variables on survival. Univariate and multivariate logistic regression was used in the analysis of post-operative blood loss and the analysis of complications. Statistica ver. 13 software (StatSoft, Cracow, Poland) was used for the analysis. A *p* value < 0.05 in all analyses was considered statistically significant.

## 3. Results

### 3.1. Group Comparison

The study group and the control group were comparable in terms of the basic sociobiological variables. There were no statistically significant differences in age, gender, weight, height, nicotinism (both active and in patients who smoked in the past), or presence of hydronephrosis (unilateral or bilateral). The waiting time for surgery from qualification during TURB to cystectomy was comparable in the laparoscopic and the open group. There were no significant differences in pre-operative WBC and creatinine values.

The nutritional status of the patients expressed by the body mass index and the laboratory results of the pre-operative albumin and total protein levels in the blood plasma were comparable in the study and control groups.

The pre-operative assessment of patients according to the ASA classification did not show any significant differences between the groups (the median ASA was 2 in both groups). Patients from the laparoscopic group had statistically significantly more comorbidity in total. They also showed a slightly lower (statistically significant) baseline hemoglobin level in the blood count. However, no statistical significance was demonstrated in terms of differences in baseline RBC and hematocrit levels between the groups.

There were no significant differences between the groups in terms of the clinical stage of the tumor and the pathomorphological assessment of the TNM scale. Also, the presence of lymph node metastases did not show any significant differences between the study and control groups. Five orthotopic reconstructions of the urinary bladder were performed in the open-label group; such operations were not performed in the laparoscopic group. The general division into urinary diversion using the intestine and simple ureterostomies showed no significant differences in this respect between the two groups of patients.

A detailed comparison of the study group and the control group in terms of the above-mentioned parameters is presented in Table 1:

### 3.2. Oncological Results

The 3 year survival after radical cystectomy is comparable in the laparoscopic and open methods. Estimated on the basis of the Kaplan–Meier curve, it is 37.7% for the study group (CI 26.9–48.5) and 44.4% for the control group (CI 33.6–55.2). The *p* long rank test for the charts of the probability of survival for both methods is 0.64, thus, showing no significant differences between the groups (Figure 1):

Multiparameter analysis taking into account the type of cystectomy (laparoscopic/open), the age of patients, BMI, the stage of the neoplastic disease, the presence of positive lymph nodes, the presence of a positive surgical margin, and the ASA score does not show differences in the probability of an overall survival between the patients in the study and the control groups (Table 2). A significant impact on the survival in this analysis is due to the advancement of the cancer (pT3-T4 tumors relative risk of death 3.02, *p* < 0.001), the presence of a positive surgical margin (relative risk of death 2.25, *p* = 0.003), and the ASA score > 2 (relative risk of death 2.06, *p* = 0.004).

The incidence of positive surgical margins in the group of patients undergoing ORC is 12.6%, while in the LRC it is 18.3%, with a *p* = 0.33. Thus, no significant differences are found in the probability of a positive margin in the test and control group.

The median number of lymph nodes removed in the study group is 11 (0–29), while in the control group it is 14.5 (0–43). The described difference is statistically significant (*p* = 0.001).

### 3.3. Blood Loss and Transfusion Rate

Patients who underwent LRC had lower intraoperative blood loss compared to the group of patients operated with the ORC method. The difference in Hgb levels before and immediately after the surgery (corrected for possible transfusion) is 1.8 mmol/L (−0.3–3.5 mmol/L) in the laparoscopy group, and 2.25 mmol/L in the ORC group (−0.5–6.36 mmol/L). The difference between these values is statistically significant (*p* < 0.001) (Figure 2).

Less intraoperative blood loss is also reflected in the less frequent need for transfusion of blood during the procedure in patients undergoing laparoscopic surgery. A total of 11.8% of patients in the laparoscopic group had packed red blood cells transfused in the operating theater, compared to 34.1% in the open group. This difference is statistically significant (*p* = 0.002) (Figure 3).

Patients in the study group also received fresh frozen plasma less frequently intraoperatively—a transfusion was performed in 4% of cases, compared to 12.2% in the control group, but this difference did not reach the assumed statistical significance (*p* = 0.110) (Figure 4).

The maximum loss of Hgb in the first three days after the surgery is significantly lower in the LRC group than in the open group: 1.8 mmol/L (−0.5–3.9) vs. 2.7 mmol/L (−0.5–4.5), *p* < 0.001 (Figure 5). Similarly, the maximum loss of RBC in the first three days after the surgery is significantly lower—in the study group, 1.03 million/µL (−0.31–2.48); in the control group, 1.42 million/µL (−0.38–2.52), *p* < 0.001.The maximum decrease in Hct in the first three days after surgery is 8.7% in the LRC group (−2.5–18.8%), and 12.7% in the ORC group (−1.8–22.4%), *p* < 0.001.

### 3.4. Complications

Statistical analysis shows no significant differences in the degree of complications assessed according to the Clavien–Dindo scale between the study and the control group within 90 days after the surgery. Both patients undergoing LRC and ORC are mostly graded at Clavien–Dindo grade 0-II (53 patients in the laparoscopic group—68.4%; 56 patients in the open group—68.3%; *p* = 0.98). Clavien–Dindo grade III-V is reported in 24 patients in laparoscopy group (31.6%), and in 26 patients from the “open” group (31.7%), *p* = 0.98. Clavien–Dindo IV-V grade in both the LRC and the ORC groups is found in 12 patients, which is 16% and 14.6%, respectively (*p* = 0.81). Death within 90 days after cystectomy was reported in 5 patients in the LRC group and 8 in the ORC group—6.6% and 9.8%, respectively (*p* = 0.66). The mean degree of complications in the Clavien–Dindo scale in the study group is 2.16 and in the control group it is 2.36, showing no significant statistical difference, *p* = 0.320 (Figure 6):

The analysis of individual complications that occurred in the majority of patients did not show any significant differences in their incidence between the groups. Patients from the LRC group required transfusion of blood products less often, which was presented in “blood loss” section. More frequent occurrence of prolonged drainage from the drain in the ORC and more frequent need for parenteral nutrition in the post-operative period in the open group is noteworthy, but in both these cases the *p* value slightly exceeds the assumed level (*p* = 0.07 and *p* = 0.06, respectively).

The detailed analysis of individual complications is presented in the table below (Table 3):

A detailed comparative analysis of individual patient subgroups shows statistically significant benefits of choosing the laparoscopic approach for some patients. Those with BMI < 30 kg/m^2^ who underwent LRC had significantly fewer inflammatory septic complications within 90 days after surgery compared to the ORC group (OR 2.6, *p* = 0.015). Similarly, fewer septic events were observed in patients with more advanced bladder cancer (pT3-T4) who underwent laparoscopic cystectomy (OR 3.65, *p* = 0.01). Significantly fewer cases of ileus and eventrations requiring surgical revision were observed in patients with an ASA score > 2 who underwent laparoscopic surgery (OR 14.0, *p* = 0.02). In patients with an ASA score < 3, a prolonged leakage from the drain was observed less frequently (OR 2.1, *p* = 0.05). In patients with cT1-T2 tumors, operated with laparoscopic technique, the need for parenteral nutrition was less frequent (OR 3.0, *p* = 0.05).

### 3.5. Other

There are no significant differences between post-operative hospitalization time in patients undergoing laparoscopic and open cystectomy. Median time from the surgery to discharging in both cases is7 days (*p* = 0.263). The operation time is slightly higher in the study group than in the control group, but the difference does not reach the assumed statistical significance (240 min vs. 225 min, *p* = 0.14). All the patients undergoing the laparoscopic and the open surgery had a liquid diet implemented on the first day after the surgery (*p* = 0.901). There was also no significant difference in time between the surgery and the implementation of full oral nutrition, which in both cases was usually applied on the fourth day after the surgery (*p* = 0.339). The duration of antibiotic treatment was also comparable in both groups and was usually 6 days (*p* = 0.807). The comparison of the above data for the study and control groups is presented in Table 4.

## 4. Discussion

Overall survival (OS) is the most important factor analyzed when assessing the effectiveness of oncological treatment. In patients with MIBC, the gold standard of treatment is the ORC [5]. Some early reports questioned the oncological results of the LRC [12]. The results presented in this study confirm the oncological safety of the laparoscopic technique in the treatment of MIBC. Other publications also indicate the equivalence of the oncological results of open and laparoscopic cystectomy. Xie et al. presented oncological results of radical cystectomy in 2098 patients from 13 centers [13]. A total of 855 patients underwent open cystectomy and 1243 had laparoscopic cystectomy (including 156 assisted by a robotic system). There were no significant differences in OS and CSS between the groups. Cox’s multi-parameter regression used by the authors also confirms the results shown in this study: stage T3-T4 tumor advancement, lymph node involvement, presence of a positive surgical margin, and ASA score > 2 are independent risk factors for patient death (HR 1.84; 2.46; 1.71, and 1.28, respectively). The choice of the laparoscopic method is not an independent predictor of the risk of death from cancer.

Two elements of the cystectomy are particularly important: the presence of a positive surgical margin and the lymph node yield (LNY). Herr et al. suggest that the positive margin ratio should be <10% and the LNY should be a minimum of 10–14 for a laparoscopic cystectomy to be considered oncologically equivalent to an open cystectomy [14]. In this study, no significant differences are found in the incidence of a positive surgical margin between the laparoscopic and the open cystectomy (18.3% and 12.6%, respectively, *p* = 0.33). LNY is lower in the laparoscopic cystectomy (11 vs. 14.5 in open cystectomy, *p* = 0.001) The publications available covering a greater number of cases indicate comparable results in terms of positive surgical margins and LNY for both surgical methods. Fonseka et al. conducted a meta-analysis comparing a total of 2104 cases of patients undergoing radical cystectomy with different surgical techniques [15]. Contrary to the present study, the authors did not find any differences in the number of lymph nodes retrieved between open and laparoscopic cystectomy. In the analyzed reports, LNY was 10–19.5 for the laparoscopic cystectomy and 10.5–18.8 for the open cystectomy, *p* = 0.99. The incidence of positive margins was 7.7% for laparoscopic cystectomy and 10.0% for open cystectomy, showing no significant differences between those methods (*p* = 0.42). Our study shows a higher percentage of positive surgical margins. However, the severity of the disease in the study population should be taken into account. In our study, half of the patients had a tumor extending beyond the bladder at the time of surgery (pT3-T4 cancer in 48.7% of patients in the laparoscopic group and 61.0% in the open cystectomy group), while in the meta-analysis by Fonseka et al., those patients constituted a definite minority (14.5% in the LRC group and 29.5% in the ORC group). This fact may explain the lower positive surgical margins ratio.

The results described in our study regarding reduced blood loss in patients undergoing LRC are similar to other reports. Unfortunately, in most cases, the publications are based on the estimated blood loss in ml, which has been shown to be a less sensitive parameter than the difference in the concentration of pre-operative and post-operative hemoglobin level [16]. Tang et al. published a meta-analysis covering 16 studies comparing laparoscopic (545 cases) and open (620 cases) cystectomy [10]. Blood loss in the laparoscopic group was reduced by an average of 467 mL (*p* < 0.001). The authors also demonstrated a lower frequency of blood transfusions in the LRC (OR 0.13%, *p* = 0.002).

In this study, no significant differences are found in the incidence and severity of early post-operative complications in patients undergoing LRC and ORC. Some groups of patients, however, have significantly fewer septic and ileus complications with LRC. Those results are in favor of the laparoscopic method, which is not inferior to the open cystectomy globally, and in individual cases it significantly reduces the risk of early surgical complications. This conclusion is also supported by other reports. Adamczyk et al. presented a significantly lower risk of complications in patients undergoing laparoscopic cystectomy, based on the mean Clavien–Dindo comparison (2.24 vs. 2.65, *p* < 0.001) [17]. The risk of complications estimated in this way reaches statistical significance in both the group of patients with simple and intestinal urinary diversion (*p* = 0.001 and *p* < 0.001). Tang et al. show a significantly lower risk of complications in patients undergoing LRC (OR 0.60, CI 0.44–0.80, *p* < 0.001) [10]. Nevertheless, there is no significant difference in the frequency of severe complications (OR 1.04, *p* = 0.86), while the benefit is significant for mild complications (OR 0.45, CI 0.33–0.62, *p* < 0.001). It should, however, be taken into account that the patients from the LRC group have a slightly lower ASA score at the beginning—the difference in the weighted mean ASA grade −0.09 (CI −0.15–0.02; *p* = 0.01). The authors also compare the risk of individual complications in more detail, showing a significantly lower risk of inflammatory septic complications in the laparoscopic group (OR 0.31; CI 0.20–0.49; *p* < 0.001) and a significantly lower risk of post-operative ileus (OR 0.54; CI 0.31–0.94; *p* = 0.03). The results described in this study show a similar trend in reducing the risk of ileus and septic complications. 

The main limitation of our study is the fact that the collected material concerns the period when the laparoscopic technique was implemented in the center for the treatment of patients with MIBC. Therefore, operators and medical staff involved in patient care were in the process of gaining experience related to learning a new method of treatment. With the increase in experience, we expect further reduction in the number of complications and improvement in oncological results in the future.

## 5. Conclusions

A radical cystectomy performed with the laparoscopic technique ensures comparable oncological results to the open approach in terms of overall survival, regardless of the stage of the disease. The percentage of positive surgical margins is comparable in the laparoscopic and the open cystectomy;The laparoscopic technique in radical cystectomy significantly reduces intraoperative and post-operative blood loss and the need for transfusion of blood products;There are no significant differences in the time of the surgery, time of post-operative hospitalization, time to the implementation of full oral alimentation between the laparoscopic, and the open cystectomy;The laparoscopic cystectomy shows no significant differences in terms of early post-operative complications compared to the open surgery. Some groups of patients, however, benefit from significantly fewer septic and ileus complications with laparoscopic cystectomy;Further studies are needed to assess the long-term outcomes of laparoscopic cystectomy for MIBC.

## Figures and Tables

**Figure 1 ijerph-19-15995-f001:**
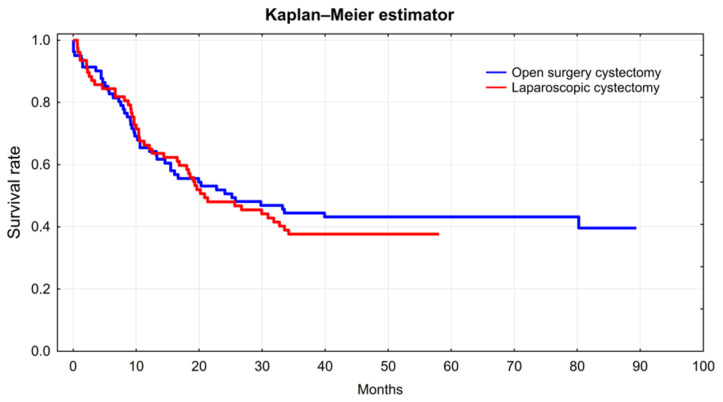
Kaplan–Meier curve LRC vs. ORC.

**Figure 2 ijerph-19-15995-f002:**
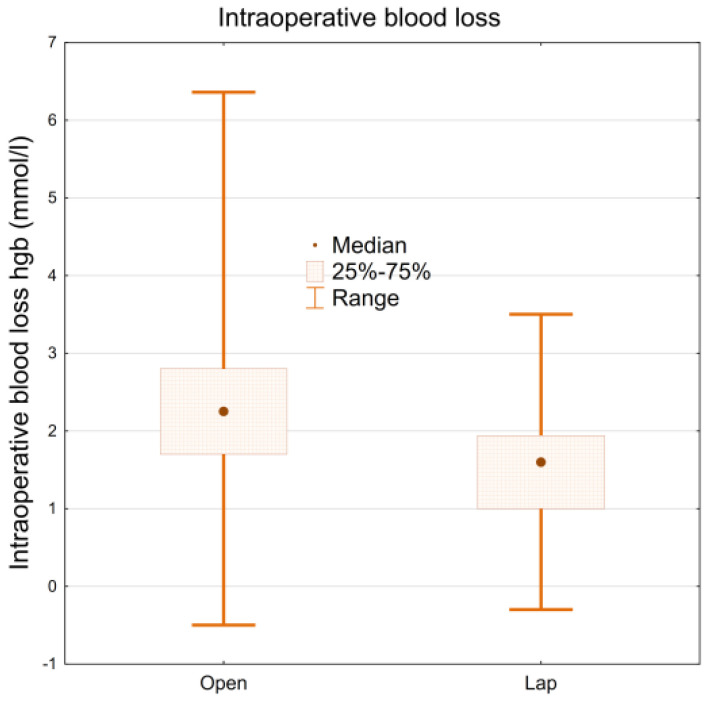
Intraoperative blood loss ORC vs. LRC.

**Figure 3 ijerph-19-15995-f003:**
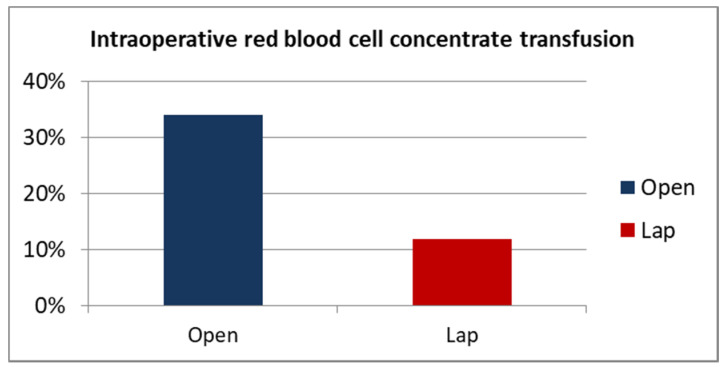
Intraoperative RBC concentrate transfusion ORC vs. LRC percentage.

**Figure 4 ijerph-19-15995-f004:**
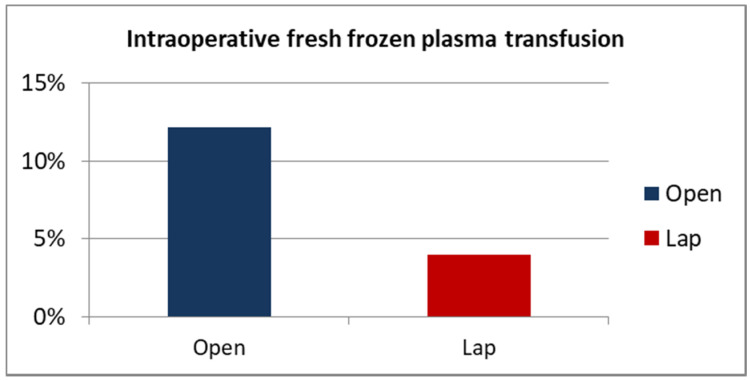
Intraoperative FFP transfusion ORC vs. LRC percentage.

**Figure 5 ijerph-19-15995-f005:**
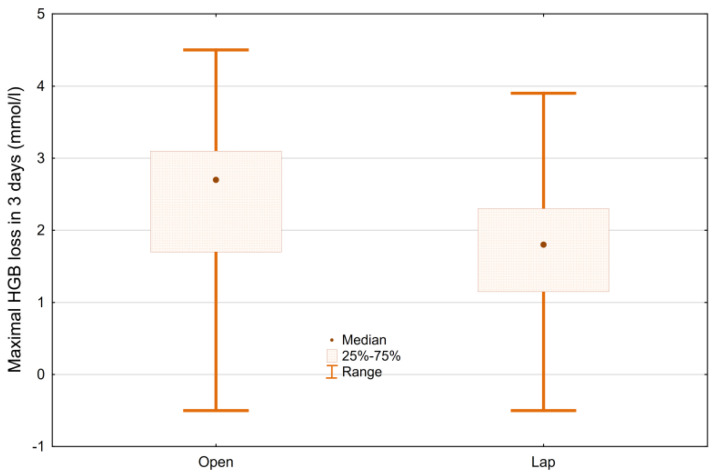
Maximal loss of HGB in 3 days after the surgery ORC vs. LRC.

**Figure 6 ijerph-19-15995-f006:**
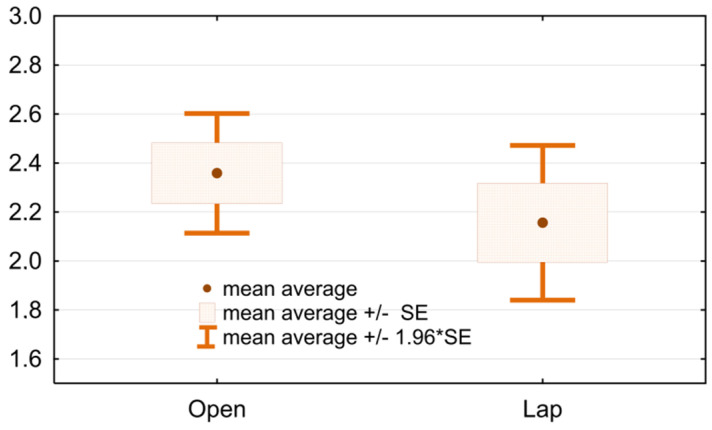
Mean degree of complications in the Clavien–Dindo scale ORC vs. LRC.

**Table 1 ijerph-19-15995-t001:** Comparison of the study and control group.

	LRC *n* = 77	ORC *n* = 82	*p*-Value
Age	66 (44–84)	65 (50–85)	0.297
Height	1.7 (1.48–1.76)	1.71 (1.5–1.89)	0.985
Weight	77 (41–120)	75 (44.5–114)	0.558
BMI	25.88 (17.34–37.02)	26.07 (17.9–34.48)	0.582
Male (sex)	62 (80.5%)	62 (75.6%)	0.450
Smoking:			
- Yes	25 (32.9)	19 (24)	0.380
- Former	44 (57.8)	49 (62)	
- Never	7 (9.2)	11 (13.9)	
Hydronephrosis			
- No	46 (60.5%)	46 (56.1%)	0.800
- Unilateral	22 (28.9%)	25 (30.5%)	
- Bilateral	8 (10.5%)	11 (13.4%)	
Waiting time for operation (days)	89 (0–510)	82 (14–208)	0.107
WBC (10^3^)	8.37 (3.01–34.02)	8.18 (3.47–32.82)	0.117
Creatinine	1.07(0.6–3.89)	0.99(0.56–8.45)	0.318
Albumin in serum	42.5 (29–49)	43 (21–49)	0.700
Total protein in serum	71 (53–78)	71 (51–78)	0.550
Hgb (mmol/L)	7.6 (5–10)	8 (4.7–10.8)	0.026
RBC (mln/µL)	4.07 (2.7–6.42)	4.33 (2.7–5.41)	0.053
Hct (%)	36.5 (24.4–48.6)	37.9 (24.1–48.2)	0.070
No. comorbidities	2 (0–6)	1 (0–8)	0.025
ASA	2 (1–4)	2 (1–3)	0.320
ASA ≥ 2	71 (93.4%)	72 (90%)	0.630
ASA ≥ 3	19 (25%)	16 (20%)	0.580
cT3–4 (operator evaluation)	35 (46%)	47 (57.3%)	0.150
pT3–4	37 (48.7%)	50 (61%)	0.120
pN+	16 (23.5%)	30 (38%)	0.060
Urinary diversion:			
- Ureterocutaneostomy	39 (51.3%)	43 (53.1%)	
- Ileal conduit (Bricker)	37 (48.7%)	33 (40.8%)	
- Studer neobladder	0	5 (6.1%)	0.070
Urinary diversion:			
- Ureterocutaneostomy	39 (51.3%)	43 (53.1%)	
- Ileal urinary diversion	37 (48.7%)	38 (46.9%)	0.820

**Table 2 ijerph-19-15995-t002:** Multiparameter analysis of ORC vs. LRC.

N = 158	Relative Risk	Standard Error	Hazard Ratio (95%) Upper	Hazard Ratio (95%) Lower	*p*-Value
Open/laparoscopic (0/1)	1.408	0.242	0.876	2.261	0.157
pT0–2: 0; pT3–4: 1	**3.024**	0.294	1.701	5.378	**<0.001**
Resection margin(0/1)	**2.25**	0.276	1.31	3.862	**0.003**
ASA ≥ 3	**2.059**	0.249	1.264	3.356	**0.004**
Sex (female 0, male 1)	0.965	0.257	0.583	1.598	0.889
Age	1.026	0.015	0.996	1.056	0.089
BMI	0.948	0.028	0.896	1.002	0.057
Neoadjuvant chemotherapy	0.962	0.303	0.531	1.744	0.899
cT0–2: 0; cT3–4: 1	1.24	0.249	0.761	2.02	0.388
N(-): 0, N(+): 1	1.718	0.303	0.949	3.11	0.074
No. of N(+)	1.055	0.053	0.952	1.17	0.309

**Table 3 ijerph-19-15995-t003:** Comparison of individual complications between LRC vs. ORC.

	LRC *n* = 77	ORC *n* = 82	*p*-Value
Clavien–Dindo > 2	24 (31.6%)	26 (31.7%)	0.98
Clavien–Dindo > 3b	12 (16%)	12 (14.6%)	0.81
Prolonged drain leak	23 (30.3%)	36 (43.9%)	0.07
Ileus (conservative treatment)	7 (9.2%)	11 (13.4%)	0.41
Need of blood transfusions	**37 (48.7%)**	**60 (73.2%)**	**0.001**
Need of parenteral nutrition	9 (11.8%)	19 (23.2%)	0.06
SIRS/sepis	22 (28.9%)	33 (40.2%)	0.13
Ileus (surgical treatment), eventration	5 (6.6%)	13 (15.9%)	0.11
Gastrointestinal hemorrhage	2 (2.63%)	2 (2.44%)	0.67
Need of PCN drainage	6 (7.9%)	5 (6.1%)	0.89
Haemodialysis	6 (7.9%)	5 (6.1%)	0.89
Thromboembolism	1 (1.32%)	3 (3.66%)	0.66
Abscess/hematoma requiring drainage	2 (2.6%)	4 (4.9%)	
Hemorrhage requiring urgent reoperation	0	1 (1.22%)	0.96
Colostomy	0	2 (2.44%)	0.51
Limb amputation	0	1 (1.22%)	0.96
ICU admission	7 (9.2%)	8 (9.8%)	0.90
Death	5 (6.5%)	8 (9.8%)	0.66

**Table 4 ijerph-19-15995-t004:** Comparison of other parameters of LRC vs. ORC.

	LRC *n* = 77	ORC *n* = 82	*p*-Value
Hospitalization (days)	7 (3–34)	7 (5–71)	0.263
Operation time (minutes)	240 (120–425)	225 (135–360)	0.140
Antibiotic therapy (days)	6 (2–27)	6 (1–70)	0.807
Time to implementation of liquid diet (days)	1 (1–6)	1 (1–4)	0.901
Time to implementation of full oral alimentation (days)	4 (2–25)	4 (2–15)	0.339

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
