# Peer review of "Comparison of Laparoscopic and Open Radical Cystectomy for Muscle-Invasive Bladder Cancer"

_ijerph, 2022, doi:10.3390/ijerph192315995_

Round 1

Reviewer 1 Report

This manuscript is entitled: “Comparison of Laparoscopic and Open Radical Cystectomy for muscle-invasive bladder cancer,” which compares laparoscopic and open radical cystectomy in the treatment of muscle-invasive bladder cancer in the Szczecin. The authors’ search for that laparoscopic cystectomy shows no significant differences in terms of early postoperative complications compared to open surgery. Overall, the proposed research is of interest with good potential. The authors carried out detailed studies to prove the concept. There are some questions and suggestions that may need to solve as below:

1.        Authors should provide an abbreviations list to make the entire manuscript easier to read.

2.        The Statistical analysis calculations data using a Shapiro-Wilk test and non-parametric U-Mann-Whitney test seem to need a quotation of the calculation formula to demonstrate the design of the geographic map.

3.        The authors should compile a table on whether other studies have reported on this statistical approach to the laparoscopic and/or open radical cystectomy for bladder cancer.

4.        Whether the data in the table are statistically relevant, such as p-value, standard deviation, etc., such data, if available, should be indicated on the table to provide more statistical confidence, such as in Figure 1.

5.          Following on from the previous question, Figures 3 and 4 also lack an assessment of the statistical meaning.

Reviewer 2 Report

The authors retrospectively investigated the difference between the patients who underwent laparoscopic and open radical cystectomy. The relationship analysis of the clinical metadata between the surgy methods are performed. However, the results are not novel. Less tissue trauma and less blood loss were reported using laparoscopic radical cystectomy. 

Reviewer 3 Report

Please do grammar check since there are missing periods, multiple spaces between words, missing prepositions, misspelled words, etc.

Discussion could include limitations of the study's data. Please discuss long-term survival of Figure 1, which seems to have significant difference between LRC and ORC.

The conclusion could include what’s the impact of current study for the field based on this study's findings, and what ideas the authors have for further research that needs to be done to advance this field.

Round 2

Reviewer 1 Report

The authors have resolved most of the concerns proposed by the reviewer, and the manuscript has been improved significantly. Therefore, we do not have further revision requirements for this updated manuscript.

Reviewer 2 Report

this manuscript can be accepted in present form